# Unattended versus Attended Blood Pressure Measurement: Relationship with Retinal Microcirculation

**DOI:** 10.3390/jcm11236966

**Published:** 2022-11-25

**Authors:** Anna Paini, Claudia Agabiti Rosei, Carolina De Ciuceis, Carlo Aggiusti, Fabio Bertacchini, Marco Cacciatore, Sara Capellini, Roberto Gatta, Paolo Malerba, Deborah Stassaldi, Damiano Rizzoni, Massimo Salvetti, Maria Lorenza Muiesan

**Affiliations:** 1Department of Clinical and Experimental Sciences, University of Brescia & Emergency Medicine ASST Spedali Civili di Brescia, 25121 Brescia, Italy; 2Department of Clinical and Experimental Sciences, University of Brescia & 2^a^ Medicina ASST Spedali Civili di Brescia, 25121 Brescia, Italy; 3Department of Clinical and Experimental Sciences, University of Brescia, 25123 Brescia, Italy; 4Department of Clinical and Experimental Sciences, University of Brescia & Medicina Montichiari ASST Spedali Civili di Brescia, 25121 Brescia, Italy

**Keywords:** unattended blood pressure, AOBP, blood pressure measurement, retinal arterioles, media/lumen ratio, wall-to-lumen ratio, adaptive optics

## Abstract

Though the relationship between both “attended” and “unattended” BP and several forms of target organ damage have been evaluated, data on retinal arteriolar alterations are lacking. The aim of our study was to evaluate the relationship between “attended” or “unattended” BP values and retinal arteriolar changes in consecutive individuals undergoing a clinical evaluation and assessment of retinal fundus at an ESH Excellence Centre. An oscillometric device programmed to perform 3 BP measurements, at 1 min intervals and after 5 min of rest was used on all individuals to measure BP with the patient alone in the room (“unattended”) or in the presence of the physician (“attended”) in the same day in a random order. The retinal arteriole’s wall thickness (WT) was measured automatically by a localization algorithm as the difference between external (ED) and internal diameter (ID) by adaptive optics (RTX-1, Imagine Eyes, Orsay, Francia). Media-to-lumen ratio (WLR) of the retinal arterioles and cross-sectional area (WCSA) of the vascular wall were calculated. Results: One-hundred-forty-two patients were examined (mean age 57 ± 12 yrs, 48% female, mean BMI 26 ± 4). Among them, 60% had hypertension (84% treated) and 11% had type 2 diabetes mellitus. Unattended systolic BP (SBP) was lower as compared to attended SBP (129 ± 14.8. vs. 122.1 ± 13.6 mmHg, *p* < 0.0001). WLR was similarly correlated with unattended and attended SBP (r = 0.281, *p* < 0.0001 and r = 0.382, *p* < 0.0001) and with unattended and attended diastolic BP (r = 0.34, *p* < 0.001 and r = 0.29, *p* < 0.0001). The differences between correlations were not statistically significant (Steiger’s *Z* test). Conclusion: The measurement of “unattended” or “attended” BP provides different values, and unattended BP is lower as compared to attended BP. In this study a similar correlation was observed between attended and unattended BP values and structural changes of retinal arterioles.

## 1. Introduction

The increase of blood pressure (BP) values is recognized as one of the most important risk factors for cardiovascular (CV) morbidity and mortality worldwide [1]. “Out of office” BP adds relevant clinical information for cardiovascular (CV) risk assessment and has reached growing relevance. However, the diagnosis and treatment of arterial hypertension are still based on the measurement of BP values in the physician’s office [2,3,4,5]. The accuracy of BP values measurement represents, therefore, an essential step in the evaluation of CV risk, due to its strong prognostic significance, as shown and confirmed in large observational studies and randomized clinical trials. In recent years growing attention has been given to self-measurement of BP, performed with the patient left alone in the room, also defined as unattended BP [4,6,7,8,9,10,11]. The unattended BP measurement has also been used in some clinical trials [8,10,11,12]. The results of studies comparing the office automated BP (OABP) measurements in the physician’s presence or absence suggest that attended BP measurement may be higher to the order of approximately 6 mmHg [8,9,10,11,12,13,14]. This difference seems to be related to changes in sympathetic nervous system activity, as indicated by a slight heart rate and skin sympathetic nerve traffic increase, and a slight muscle sympathetic nerve traffic decrease [15].

As previously reported by our group [16] and suggested also by other authors [8,17,18,19], BP values recorded by the “attended” or “unattended” approach are not equivalent, even when obtained with the same oscillometric device [10,11]. This observation has limited the clinical significance of findings from clinical trials such as the SPRINT study [10,11].

Moreover, it remains to be established whether unattended BP values may more accurately predict the risk of future CV events at the individual patient level [4,11,12,20,21,22]. The relationship between unattended BP and the presence of organ damage, which represents a subclinical condition strictly related to CV risk, has been evaluated in few studies [16,22,23,24,25]. All these studies have demonstrated that attended and unattended BP are similarly related to left ventricular mass index and relative wall thickness [24], as well as to carotid artery intima-media thickness or plaque [22,25] and to aortic stiffness [25].

Microvascular structural changes also represent an early marker of CV disease, providing reliable prognostic information. Structural alterations in the microcirculation may lead to impairment of organ flow reserve and to a faster progression of organ damage and the increased CV risk of diabetic and hypertensive patients [26,27].

The vascular damage in the retinal microcirculation may reflect the increase in the media or the total wall thickness to internal lumen diameter ratio (i.e., media-to-lumen ratio, MLR or wall to lumen ratio, WLR) in small resistance arteries occurring in other districts [28], often accompanied by endothelial dysfunction. Recently, scanning laser Doppler flowmetry and adaptive optics have been proposed as non-invasive methods to extend the microvasculature evaluation to a broader range of patients and clinical settings and are increasingly used to estimate the WLR of retinal arterioles [29]. In only one study was unattended automated office systolic BP correlated significantly with structural parameters of retinal microcirculation, in particular WLR and wall thickness, assessed by scanning laser Doppler flowmetry [30]. No data are available on the relationship between retinal structural alterations assessed by adaptive optics and both unattended and attended BP measurements in individuals with different cardiovascular risk.

Therefore, we thought it worthwhile to evaluate the relationship between WLR of the retinal arteries measured by means of the adaptive optics technique and BP values obtained by both the attended (i.e., measured by the physician) and the “unattended” (i.e., patient alone in the room) approaches, in the same day and in the same room, using the same device (automated oscillometric technique) [16].

## 2. Methods

Our institutional guidelines procedures were followed. Each patient gave informed consent to participate in the study. The protocol of the study followed the ethical guidelines of 1975 the Declaration of Helsinki, and was approved by the institution’s Human Research Committee. The corresponding author is available to share the data collected for this study upon reasonable request. No animals were involved in this study.

One-hundred-forty-two consecutive outpatients undergoing a diagnostic work-up for known or suspected arterial hypertension were prospectively enrolled into the study. All included patients were at the European Society of Hypertension (ESH) Excellence Centre in Brescia (Italy) from February 2017 to December 2019. A thorough clinical examination, including anthropometric measurements, was performed in all patients. A careful assessment of CV risk factors including the presence of diabetes and dyslipidemia, smoking status and a documented clinical history was also obtained for each individual. All individuals refrained from alcohol, caffeine-containing beverages, and cigarettes in the 3 h prior to the visit and were in a fasting state. All BP measurements and evaluation of retinal parameters were performed on the same day.

### 2.1. Blood Pressure Measurement

BP values were measured according to a standardized protocol by a physician in a room with controlled temperature at 22 ± 1 °C. The cuff and bladder dimensions were adapted to patients’ upper arm circumferences and the automated oscillometric device Omron HEM 9000Ain was used [16,31]. All patients were left alone and the device was programmed by the physician to perform, after 5 min of rest, 3 automatic BP measurements with 1 min intervals for the measurement of “unattended” BP. Attended BP measurements were obtained by manual activation of the device performed by the physician sitting in front of the patient; after 5 min of rest 3 BP automated measurements at 1 min intervals were carried out; the physician did not interact with the patient during this time interval. Systolic (SBP) and diastolic (DBP) blood pressure were calculated as the mean of three consecutive readings for both unattended and attended BP measurements [16]. The unattended and attended measurements were performed in a random order. Both measurements were performed in alternating sequence in successive participants (attended first and unattended afterward in one patient, and vice versa for the subsequent patient).

### 2.2. Evaluation of the Retinal Microcirculation

Retinal arteries were evaluated with adaptive optics technique using the Rtx-1 optical camera (Imagine Eyes, Orsay, France), according to a previously specified protocol [29,32,33]. The measurements were taken with the patient in a sitting position, after 5 min of rest. The superior temporal portion of the optic disc of the right retina was examined. The internal and external diameter of the arterioles were measured and then wall cross sectional area (WCSA) and WLR ratio were calculated with a specific software [32,33,34,35].

### 2.3. Statistical Analysis

All values represent means and SD, or percentages. A *p* < 0.05 was considered significant. The relationship between variables was assessed by linear correlation and the Pearson’s correlation coefficients were measured. The correlation coefficients of BP values with WCSA and WLR ratio were compared by Steiger’s Z statistic. The area under the curve (AUC) to estimate the predictive power of attended and unattended BP values for the presence of an increased WLR (defined as the value above the median value, identified as 0.27 µm)was calculated. The AUCs were compared by the De Long method.

All analyses were performed with IBM SPSS software (version 23; SPSS Inc., Chicago, IL, USA) and MedCalc for Windows, version 15.0 (MedCalc Software, Ostend, Belgium).

### 2.4. Results

Table 1 shows the mean age and prevalence of gender and of CV risk factors of all patients included in the study. Among hypertensives (60%) 84% were treated, with different antihypertensive drugs. The prevalence of dyslipidemia and diabetes were 34 and 11%, respectively, while 12% of individuals were active smokers.

Both systolic and diastolic unattended BP were lower as compared to attended BP (SBP: 129 ± 14.8 vs. 122.1 ± 13.6 mmHg, *p* = 0.0001; DBP: 76.1 ± 9.4 vs. 74.3 ± 8.5 mmHg, *p* = 0.0001). The mean differences of values obtained by the two approaches were 6.9 ± 7.5 mmHg for SBP and 1.8 ± 4.5 mmHg for DBP (median: 6.0 and 2.0 mmHg, respectively). The Spearman’s coefficient of correlation between the two methods of BP measurement was r = 0.864 (*p* < 0.0001) for systolic values and r = 0.880 (*p* < 0.0001) for diastolic values.

### 2.5. Vascular Organ Damage

The mean values of WCSA and of WLR were 4305.7 ± 943 µm^2^ and 0.27 ± 0.04, respectively (Table 2).

We observed a positive statistically significant correlation between WLR and both attended SBP (r = 0.281, *p* < 0.0001) and unattended SBP (r = 0.38, *p* < 0.0001) (Figure 1) and both attended and unattended DBP (r = 0.34 r = 0.29, all *p* < 0.001). A significant correlation was also observed between WLR and both attended and unattended mean BP (r = 0.35, *p* < 0.0001 r = 0.377 *p* < 0.0001). (Figure 2). These correlations remained significant after adjustment of WLR for age and sex without significant differences between attended and unattended values (attended SBP r = 0.29, *p* < 0.001; unattended SBP r = 0.38, *p* < 0.0001; attended diastolic BP r = 0.32, *p* < 0.0001 and unattended diastolic BP r = 0.28 *p* < 0.001). No difference in the correlation coefficients was observed between attended and unattended systolic, diastolic and mean pressure values and both unadjusted and sex and age adjusted WLR (Steiger’s Z-test *p* = ns).

We built ROC curves for attended and unattended SBP to predict the presence of an increased WLR.

No significant differences between the discrimination value of attended or unattended BP for the detection of increased WLR were reported (attended SBP: AUC 0.597, 95% CI 0.511–0.678 vs. unattended SBP: AUC 0.641, 95% CI 0.556–0.719, *p* for the comparison = ns).

ROC curve analysis showed that the optimal cut-point for attended or unattended SBP to discriminate the presence of an increased WLR was 139 mmHg for attended SBP and 124 mmHg for unattended SBP. A value greater than 139 mmHg for attended SBP had a 31% sensitivity and a 90% specificity for the prediction of an increased WLR, while a value greater than 121 mmHg for unattended BP had a 50% sensitivity and 82% specificity for the prediction of an increased WLR.

No correlations between WLR and other variables (age, sex, BMI, heart rate, estimated glomerular filtration rate) were observed. We assessed the correlations in patients untreated and treated with antihypertensive drugs and observed significant correlations between WLR and attended and unattended BP, without significant differences between attended and unattended values in untreated patients (attended SBP r = 23, *p* = 0.03; unattended SBP r = 0.38, *p* < 0.0001; attended diastolic BP r = 0.27, *p* = 0.008 and unattended diastolic BP r = 0.26 *p* = 0.014) and in those receiving treatment (attended SBP r = 44, *p* = 0.002; unattended SBP r = 0.38, *p* < 0.001; attended diastolic BP r = 0.41, *p* = 0.003 and unattended diastolic BP r = 0.34 *p* =0.014).

In addition, no significant correlations were observed between WCSA and demographic, anthropometric or hemodynamic variables, including attended or unattended BP.

## 3. Discussion

In this study, we present the relationship between the two attended and unattended approaches for BP measurement and structural changes in the retinal arterioles assessed by the adaptive optics technique. To our knowledge, this is the first finding obtained by the two BP measurements in individuals undergoing the retinal fundus examination by adaptive optics technique. Our results show that unattended and attended BP are similarly correlated to WLR.

Few studies have tried to analyze the correlation of unattended BP with hypertension mediated organ damage [22,23,24,25], focusing instead on left ventricular mass [23,24,25], intima-media thickness or plaque [22,23] or pulse wave velocity [25], with some conflicting results.

The discrepancy between the studies might be accounted for by several aspects, including a slight difference between “unattended” and “attended” BP, a single-center design, and by the characteristics of the patients, including diabetes, dyslipidemia, rheumatoid arthritis [35,36] and use of antihypertensive/statins treatment.

Only one study [30] has investigated and found a significant correlation between automated office systolic BP and WLR, reflecting the structural alterations in microcirculation. It should be also underlined that the study was aimed at establishing the values of WLR in a reference population, i.e., within the range of normal BP values, and used another technique, i.e., laser scanner Doppler flowmetry in a group of 256 individuals free from clinically evident CV disease, diabetes and drug treatment for hypertension and/or dyslipidemia. The authors found no significant relationship between office SBP and all retinal parameters, while automated office SBP, after adjusting for age, reached a better correlation with WLR than office SBP, thus confirming a significant association between perhaps the earliest form of target organ damage and BP.

Structural and functional alterations of small arterial vessels are commonly associated with physiological aging. Small resistance arteries also show an increase in MLR or WLR [26,27,34,35], as well as macrovascular alterations [36,37,38], in hypertension, diabetes, and obesity. Micromyography allows the measurement of MLR or WLR of arterioles obtained from subcutaneous tissue biopsies and has been considered the “gold standard” method for assessing microvascular structural alterations [34], with a strong prognostic significance. Scanning laser Doppler flowmetry and adaptive optics have been more recently introduced and, because of their non-invasive nature, are increasingly used for estimating the WLR of retinal arterioles in a broader range of patients and clinical settings [29,34]. In the present study we observed a tighter correlation between WLR of retinal arterioles and both attended and unattended office BP. Our group previously observed a correlation between WLR of retinal arterioles assessed by scanning laser Doppler flowmetry and clinic or 24-h blood pressures [39]. The evaluation of the retinal arteriolar morphology by adaptive optics may provide a more precise assessment of microvessel morphology, with similar information as compared with the invasive micromyographic measurement of the MLR of subcutaneous small arteries [33]. In the present study attended and unattended office BP were significantly correlated with the WLR of retinal arterioles, but not with other morphological parameters, including internal or external diameter or wall cross-sectional area. This is not surprising, since WLR is considered to be the most informative parameter in terms of evaluation of retinal arteriolar morphological changes. In fact, in a previous study [40] only WLR and wall thickness was increased in hypertensive subjects compared with normotensive subjects while internal or external arteriolar diameters were not significantly different. It is probable, although it has not been directly demonstrated, that, as previously observed for MLR of subcutaneous small resistance arteries [35], WLR may be in large part independent of vessels’ dimension, being therefore a most reliable indicator of microvascular damage [41].

In this study we have observed a difference of 6 mmHg between the attended and unattended BP mean values, which is comparable to the results previously reported by all studies with ‘research quality’ BP measurements. In other studies, smaller or larger differences have been observed, possibly related to the use of the auscultatory method for BP measurement, with a different number of measurements considered, and in a real-life setting [14].

The measurement of unattended BP was initially proposed by Myers et al. [42] with the aim of removing the white coat effect. Thereafter it was indicated as the preferred method of performing in-office BP measurement by the 2016 and 2018 Canadian Hypertension Guidelines [4].

The AHA 2017 Hypertension Guidelines [3] underlined some potential benefits of unattended BP measurement in order to avoid the white coat effect, without giving a specific option for AOBP. Among the theoretical advantages of the unattended approach, the lack of patients’ interaction may reduce anxiety and offer a better standardization of BP measurement procedure, giving values that are more closely correlated with daytime monitored BP [14,43] and home BP. The prognostic significance of AOBP has been reported in very few studies, so far [44].

Our results have added information on the relationship between subclinical organ damage and unattended BP measurement. Preclinical target organ damage represents an intermediate step between the increase in BP values and other CV risk factors and the development of clinical CV complications. Hypertension guidelines underscore the role of subclinical OD assessment for a more accurate and precise CV risk stratification [2,3]. In addition, the reduction of CV events is closely associated with changes (reduction of less progression) of preclinical OD occurring during treatment and may reflect the efficacy of antihypertensive treatment. The non-invasive evaluation of microvasculature could improve the role of microvascular function and structure, being probably the earliest phenotype of hypertension related organ damage.

### Limitations

Our results should be considered within the context of its limitations. The study population consisted of Caucasian individuals (99.5%), free from CV disease, with a diabetes mellitus prevalence of 11% and cannot be extended to individuals with different CV risk profiles.

In this study a physician performed attended BP measurements. The relationship between preclinical hypertension mediated organ damage and unattended and/or attended BP values measured by non-physicians could be addressed in the future. Furthermore, we did not measure BP by the auscultatory method.

The main limitation for a widespread use of non-invasive techniques for retinal arterioles measurement is the lack of prognostic significance [29], although future studies are ongoing with the aim of evaluating the incidence of CV events that are associated with an increase of WLR, confirming previous results obtained by the micromyographic technique. If this is the case, the non-invasive assessment of the microvascular function and structure will definitively move from bench to bedside [29].

Our study was cross sectional and the relationship between changes in attended or unattended BP and changes in retinal microvasculature and/or of other forms of preclinical organ damage remains a matter of future investigations.

## 4. Conclusions

Our results suggest that BP values measured by the automated oscillometric technique, either in the presence or in the absence of the physician and performed according to a research accurate protocol, are equally correlated to retinal microvasculature structure. In addition, the presence of an increased WLR could correspond to different BP values, according to the method used for measurement and in the presence or absence of a physician.

## Disclosures

This manuscript was presented in an abstract form at the National Meeting of Italian Society of Hypertension in 2021.

## Figures and Tables

**Figure 1 jcm-11-06966-f001:**
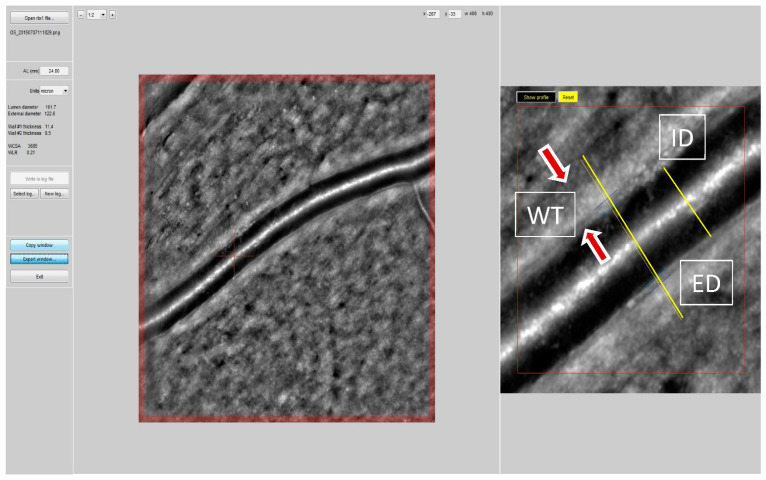
An example of retinal arteriole’s lumen and wall thickness measurement. ID internal diameter; ED external diameter; WT wall thickness.

**Figure 2 jcm-11-06966-f002:**
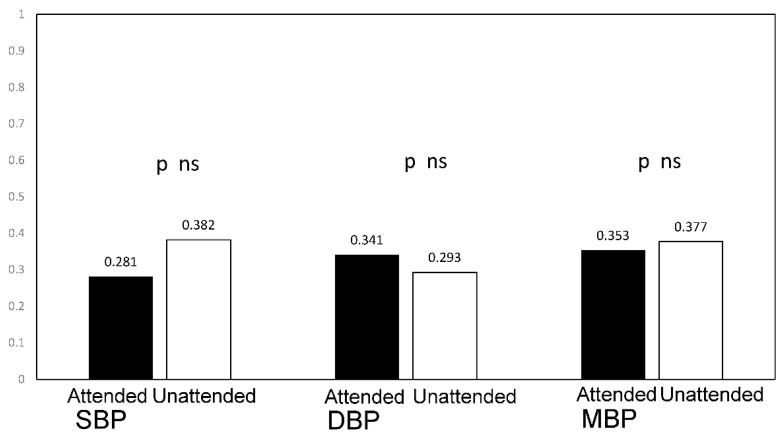
Correlation between attended and unattended SBP, DBP and MBP values and wall-to-lumen ratio (WLR). SBP systolic blood pressure; DBP diastolic blood pressure; MBP mean diastolic blood pressure.

**Table 1 jcm-11-06966-t001:** Demographic and clinical characteristics of the population.

Variables	N = 142 Patients
Age (years)	57 ± 12
Sex (males/females) (%)	334/230 (59%, 41%)
Height (cm)	169 ± 9
Weight (kg)	75 ± 16
BMI (kg/m^2^)	26 ± 4
Hypertension, *n* (%)	85 (60%)
Antihypertensive treatment (%)	71 (84%)
Diuretics, *n* (%)	17 (12%)
β-blockers, *n* (%)	25 (18%)
CC-blockers, *n* (%)	29 (21%)
ACE-i or ARB, *n* (%)	26 (20%)
Potassium-sparing diuretics, *n* (%)	27 (19%)
α-blockers, *n* (%)	11 (8%)
Dyslipidaemia, *n* (%)	84 (34)
Diabetes, *n* (%)	34 (11)
Smoking (yes) (%)	17/80/45 (12/56/32)

CC-blockers: Dihydropyridinic calcium channel blockers; ACE-i: Angiotensin-converting enzyme inhibitors; ARB: Angiotensin receptor blockers.

**Table 2 jcm-11-06966-t002:** Retinal arterioles parameters measured by adaptive optics technique.

	Mean Value	Standard Deviation
Internal lumen diameter (µm)	94.27	12.8
External diameter (µm)	119.7	14.7
WCSA (µm^2^)	4305.7	943.1
Wall-to-lumen ratio (WLR)	0.27	0.04

## Data Availability

The data presented in this study are available on request from the corresponding author.

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
