# Peer review of "Unattended versus Attended Blood Pressure Measurement: Relationship with Retinal Microcirculation"

_jcm, 2022, doi:10.3390/jcm11236966_

Round 1

Reviewer 1 Report

Dr Anna Paini evaluated the relationship between attended or unattended blood pressure value and organi damage such as retinal arteriolar changes. The work is well done and the results are important, scientifically tested. Research area is extremely important and deserve investigationI recommend that the authors review the English language and correct some imperfections that the manuscript presents.

Author Response

We thank the reviewers for their thoughtful comments and respectfully resubmit our revised paper.

We have carried out the required revisions, and we believe that the paper is improved.

Please find a point by point description of our responses to their comments.

Reviewer #1:

Reply We wish to thank you very much for the appreciation of our work. We hope that the changes made according to reviewers 2 and 3 may further improve the quality of the manuscript. We apologize for errors in English language and tried to revise english language

Reviewer 2 Report

To evaluate the relationship between “attended” or “unattended” blood pressure values and retinal arteriolar changes in consecutive individuals undergoing a clinical evaluation and assessment of retinal funds, this author made a cross-sectional study of 142 patients. From this study, this author concluded that the measurement of “unattended” or “attended” blood pressure provides different values. However, present study contains many critical problems. I commented those critical points to this author as following.

1) Abbreviation should not use in title. The word “BP” should be spelled out.

2) The abstract is redundant. The background and method paragraph in abstract is too long. 

3) The conclusions could not be supported by the results of this study. Even this author concluded as measurement of “unattended” or “attended” blood pressure provides different values, there were no significant difference between those two values.

4) In abstract section, there is a description as “r=0.281 e r=0.382, p<0.0001). Why there are two values of r but not for p?

5) Align the number of digits after the decimal point is necessary. There is a description as “r=0.281 e r=0.382” and “r=0.34 e r=0.29).

6) Without an annotation in “r” this author used “r”. Is this “r” means simple correlation coefficient or partial correlation coefficient?

7) All results of present study are calculated only by unadjusted model. Since the range of present study population is wide (±12 years), age could influence on present results. In addition to that, sex differences and menopausal status also could influence on present results. Careful consideration for the confounders on present results should be performed. Renal function also could influence on present results.

8) Without describing the definition of increased WLR, this author performed ROC curves that evaluated the optimal cut-point for systolic blood pressure.

9) To evaluate the association between increased WLR and blood pressure, odds ratios and 95% confidence intervals of increased WLR for targeted variables should be calculated.

10) In this study, no correlations were observed between WLR and known variables such as age, sex, BMI, heat rate. Those results also raise another problem. How is the representativeness of the target population in this study? The results that showed no correlations among those variables were compatible with other previous studies?

11) In there any evidence that showed present measurement values of WLR are appropriate to evaluate retinal microcirculation? Since no correlations were observed between WLR and known variables, the values of WLR itself became doubtful. How is the correlation between WLR and estimated glomerular filtration rate?

12) How is the influence of medication on present results? Only by crude model, this author made conclusion. This is not an appropriate manner to make conclusion.

Author Response

Reviewer #2

Reply We wish to thank you very much for your comments and suggestions. We hope that the changes made in response have improved the quality of the manuscript.

  • Abbreviation should not use in title. The word “BP” should be spelled out.

Reply Thank you for this comment.  We made the correction

  • The abstract is redundant. The background and method paragraph in abstract is too long. 

Reply The background and methods paragraphs have been shortened

  • The conclusions could not be supported by the results of this study. Even this author concluded as measurement of “unattended” or “attended” blood pressure provides different values, there were no significant difference between those two values.

Reply We apologize for the mistake  The statistical significance of differences have been now reported

  • In abstract section, there is a description as “r=0.281 e r=0.382, p<0.0001). Why there are two values of r but not for p?

Reply  The p value was the same, we have now given the p value for each correlation

  • Align the number of digits after the decimal pointis necessary. There is a description as “r=0.281 e r=0.382” and “r=0.34 e r=0.29).

Reply  we have followed the suggestion, thank you

  • Without an annotation in “r” this author used “r”. Is this “r” means simple correlation coefficient or partial correlation coefficient?

Reply  R means  Pearson simple correlation coefficient  (this poin has been clarified in the statistical analysis section)

  • All results of present study are calculated only by unadjusted model. Since the range of present study population is wide (±12 years), age could influence on present results. In addition to that, sex differences and menopausal status also could influence on present results. Careful consideration for the confounders on present results should be performed. Renal function also could influence on present results.

Reply  We agree that several confounders may influence the difference between attended and unattended BP. However the aim of our analysis was to assess whether in the same individuals WLR could be related to attended or unattended BP to a different degree. We have performed the analysis by using age and sex adjustment

  • Without describing the definition of increased WLR, this author performed ROC curves that evaluated the optimal cut-point for systolic blood pressure.

Reply The definition of increased WLR is reported in the method section (value above the median value, identified as 0.27 µm). 

  • To evaluate the association between increased WLR and blood pressure, odds ratios and 95% confidence intervals of increased WLR for targeted variables should be calculated.

Reply Thank you for this comment , we have reported odds ratios and 95% confidence intervals of increased WLR for targeted variables

  • In this study, no correlations were observed between WLR and known variables such as age, sex, BMI, heat rate. Those results also raise another problem. How is the representativeness of the target population in this study? The results that showed no correlations among those variables were compatible with other previous studies?

Reply  Thanks for raising this important issue. We have not found correlations with the variables you suggest, as well as other studies performed using  the AO technique. In the study by  Meixner E, Michelson G a correlation with age was observed in the younger group of subjects, aged 20 to 40 years, the difference in WLR between those aged  40 to 59 and those over 60 years was not significant.   We believe that the different mean age (57±12 years in our study) might have influenced the lack of correlation between WLR and age.( Meixner E, Michelson G. Measurement of retinal wall-to-lumen ratio by adaptive optics retinal camera: a clinical research.  Graefes Arch Clin Exp Ophthalmol. 2015. PMID: 26267750)

As far as BMI is concerned, Meixner  & Michelson described an increase in WLR in individuals with  obesity, and Grassi et al have reported the same results in severe obese individuals by the micromyographic technique. On the opposite no changes in WLR were observed in dyslipidemic or obese patients by Rosenbaum et al (Rosenbaum D, Mattina A, Koch E, Rossant F, Gallo A, Kachenoura N, Paques M, Redheuil A, Girerd X. Effects of age, blood pressure and antihypertensive treatments on retinal arterioles remodeling assessed by adaptive optics. J Hypertens. 2016. PMID: 27065002)

Nevertheless these correlations remained significant after adjustment of WLR for age and sex, without significant differences between attended and unattended values (attended SBP r=0.29, p<0.001; unattended SBP r=0.38, p<0.0001; attended diastolic BP r=0.32, p<0.0001 and unattended diastolic BP r=0.28 P<0.001). No difference in the correlation coefficients was observed between attended and unattended systolic, diastolic and mean pressure values and both unadjusted and sex and age-adjusted WLR (Steiger’s Z-test p=ns).

  • In there any evidence that showed present measurement values of WLR are appropriate to evaluate retinal microcirculation? Since no correlations were observed between WLR and known variables, the values of WLR itself became doubtful. How is the correlation between WLR and estimated glomerular filtration rate?

Reply  Thanks for your comment . As recently pointed out in this review  WLR evaluated by AO shows a linear relationship with age and blood pressure values, and may be reduced  by the use of antihypertensive drugs.  The  MLR of subcutaneous small resistance arteries measured by wire micromyography was compared with WLR evaluated by either AO and the evaluation of the WLR by AO showed a clear advantage over that using laser Doppler flow technique This new reference was added as reference 32  (Rizzoni D et al A New Noninvasive Methods to Evaluate Microvascular Structure and Function.Hypertension. 2022 May;79(5):874-886.)

Previous studies did not find a correlation between WLR by AO technique and eGFR , but with albuminuria . We did not find a significant correlation between WLR by AO technique and eGFR

  • How is the influence of medication on present results? Only by crude model, this author made conclusion. This is not an appropriate manner to make conclusion.

Reply  Thanks for your important comment.We assessed the correlations in patients untreated and treated with antihypertensive drugs . Significant correlations in both groups between WLR and attended and unattended BP were observed , without significant differences between attended and unattended values in untreated patients (attended SBP r=23, p=0.03; unattended SBP r=0.38, p<0.0001; attended diastolic BP r=0.27, p=0.008 and unattended diastolic BP r=0.26 p=0.014) and in those receiving treatment (attended SBP r=44, p=0.002; unattended SBP r=0.38, p<0.001; attended diastolic BP r=0.41, p=0.003and unattended diastolic BP r=0.34 p =0.014)

Reviewer 3 Report

Presented paper is prepared by the authors of the paper published in Hypertension in 2019:   Salvetti M, Paini A, Aggiusti C, Bertacchini F, Stassaldi D, Capellini S, De Ciuceis C, Rizzoni D, Gatta R, Agabiti 392 Rosei E, Muiesan ML. Unattended Versus Attended Blood Pressure Measurement. Hypertension. 2019 393 Mar;73(3):736-742.

The current analysis is similar, but authors analyze relation of attended and unattended  blood pressure measures and media to lumen ratio (WLR) of the retinal arterioles.

As found in previous report concerning LVH and IMT,  there is no difference in relations of both types of BP measurement and WLR.  Study is performed in similar methodological approach as previous one, using the same statistical measures – that is the strength of the study.  The method to compare AUC should be provided - DeLong method, other?

Authors should  explain clearly, why do they think that WLR relation with both types of BP measurement is assumed to be different from LVH and IMT.

Authors should present what is clinical impact of the study.

Author Response

Reviewer #3

.

As found in previous report concerning LVH and IMT, there is no difference in relations of both types of BP measurement and WLR.  Study is performed in similar methodological approach as previous one, using the same statistical measures – that is the strength of the study.  The method to compare AUC should be provided - DeLong method, other?

Reply Thank you for this comment, we apologize and now report that the AUC were compared by the De Long method

Authors should  explain clearly, why do they think that WLR relation with both types of BP measurement is assumed to be different from LVH and IMT.

Reply Thank you for this important comment  Cardiac mass, carotid IMT and small arteries may have different determinants, obviously including BP.  We did not assume that  WLR relation with both types of BP measurement was different from the one of LVH and IMT. We simply wanted to verify whether attended and unattended BP relationship was similar to the one of other organ damage phenotype

Authors should present what is clinical impact of the study.

Reply Thank you for this important comment .  From a clinical point of view the practical importance is that the presence of an increased WLR could correspond to different BP values, according to the method used for measurement, in the presence or not of the physician .

Round 2

Reviewer 2 Report

I checked this revised version of manuscript. I found this author successfully solved all my concerns. Now I don’t have any concern.

Reviewer 3 Report

Authors did respond to all queries. Paper is written properly, methods were chosen and applied correctly.